# Daily Adequate Intake of Fruit and Vegetables and All-Cause, Cardiovascular Disease, and Cancer Mortalities in Malaysian Population: A Retrospective Cohort Study

**DOI:** 10.3390/nu16183200

**Published:** 2024-09-21

**Authors:** Lay Kim Tan, Nabilah Hanis Zainuddin, Najjah Tohar, Ridwan Sanaudi, Yong Kang Cheah, Mohd Azahadi Omar, Chee Cheong Kee

**Affiliations:** 1Sector for Biostatistics & Data Repository, Office of NIH Manager, National Institutes of Health, Ministry of Health Malaysia, Shah Alam 40170, Selangor, Malaysia; drnabilah.h@moh.gov.my (N.H.Z.); najjahtohar@moh.gov.my (N.T.); ridwan.s@moh.gov.my (R.S.); drazahadi@moh.gov.my (M.A.O.); kee@moh.gov.my (C.C.K.); 2School of Economics, Finance and Banking, College of Business, Universiti Utara Malaysia, Sintok 06010, Kedah, Malaysia; yong@uum.edu.my

**Keywords:** daily adequate fruit and vegetable intake, all-cause mortality, CVD mortality, cancer mortality, Malaysian population

## Abstract

Background/Objective: We investigated the relationship between daily adequate FV intake and risk of all-cause, cardiovascular disease (CVD), and cancer mortalities among Malaysian adults. Methods: Data from a total of 18,211 Malaysian adults aged 18 years and above whom participated in the National Health and Morbidity Survey 2011 were analyzed. The participants were followed up for approximately 11 years, and mortality data were ascertained through record linkages with the death registry from the Malaysian National Registration Department. Multiple Cox regression was applied to assess the association between daily adequate FV intake and risk of all-cause, CVD, and cancer mortalities, adjusting for sociodemographic, lifestyle, and health conditions. Results: During the follow-up period, we observed a total of 1809 all-cause, 374 CVD, and 216 cancer mortalities. No significant association between daily adequate FV intake with all-cause mortality (adjusted hazard ratio (aHR): 1.01, 95% confidence interval (CI: 0.79–1.31), CVD mortality (aHR: 0.91, 95% CI: 0.57–1.47), and cancer mortality (aHR: 1.27, 95% CI: 0.74–2.17) were observed, even after excluding deaths that occurred in the first two years of observation. Conclusions: Further investigation on the type of FV intake and its preparation method with risk of mortality will provide a holistic insight into the causal relationship between FV intake and mortality.

## 1. Introduction

Fruit and vegetables (FVs) are universally promoted as healthy and essential for human nutrition due to their substantial proportion of vitamins (e.g., vitamins A and C), minerals (e.g., electrolytes), and phytochemicals (e.g., antioxidants), as well as source of dietary fiber. Hence, daily adequate intake of FVs has been recognized as an important component of a healthy diet and has also become a key component of multiple national dietary guidelines. Most nutritional and global recommendations include the consumption of at least two servings of fruits and three servings of vegetables per day for adults [1,2]. Malaysia contributed to this global effort of the healthy eating campaign where the FV group in the Malaysia Food Pyramid was switched with the grains group to the pyramid base (i.e., level 1) in the year 2020, which further emphasizes the importance of an optimal number of FV servings [3].

Besides the basic nutrition needs in humans, FVs have potential health-promoting effects, including their role in reducing inflammation and their potential preventive effects on various chronic diseases. Evidence from clinical and observational epidemiological studies has shown inverse associations between daily adequate intake of FVs and chronic diseases, including various types of cancers, cardiovascular diseases (CVDs), and neurodegenerative diseases. A systematic review and dose–response meta-analysis of prospective studies that included a total of 142 publications from 95 unique cohort studies, where forty-four studies were from Europe, twenty-six were from the USA, twenty from Asia, and five from Australia, reported that adequate intake of FVs was associated with reduced risk of total cancer with the significant relative risk of 0.97 [4]. Likewise, multiple systematic reviews assessing FV intake and various CVD outcomes in various populations demonstrated the same direction of association, where the daily adequate intake of FVs was inversely associated with the risk of CVD [5]. Previously, we have reported that daily adequate intake of FVs has a 23% lower risk for the respective hypertension and hypercholesterolemia conditions among Malaysian adults who were not aware of having these health conditions [6].

Epidemiological analyses showed that daily adequate intake of FVs could reduce the risk of mortality. For instance, a Chinese nationwide population-based study, which sampled more than half a million adults aged between 30 and 79 years old, concluded that fruit intake was associated with a significant 34% lower CVD mortality, 17% lower cancer mortality, and 42% lower mortality from chronic obstructive pulmonary disease [7]. Recently, the findings from two prospective cohort studies and a meta-analysis of 26 cohort studies involving men and women from the United States (US) demonstrated that adequate daily intake of FVs was associated with 13% lower total mortality and 12%, 10%, and 35% lower mortalities for CVD mortality, cancer mortality, and respiratory disease mortality, respectively [8]. A similar trend of association was reported in the Japanese and European populations, with a significant 24% and 3% lower CVD mortalities, respectively, among individuals with adequate FV intake [4,9]. These findings were, however, inconsistent when no significant differences in risk of mortality were observed between vegetarians and non-vegetarians in a British population [10,11], as well as no association between FV intake and cancer mortality reported in large prospective studies [12,13,14,15]. The inconsistency of these findings further warrants investigation in other populations, including the Malaysian population.

Despite the continuous effort in preventive healthcare education by healthcare professionals emphasizing the importance of healthy dietary patterns, including the daily adequate intake of FVs, Kee et al. have reported a significant declining trend in daily adequate intake of FVs among Malaysian adults over a 13-year period (i.e., 2006 to 2019) from 3.9% to 2.3% [16]. To date, there are no data about the relationship between daily adequate FV intake and mortality risk among Malaysian adults. These data will not only encourage consumers to have healthier choices of food but also aid the public health authorities and/or policymakers to further improvise the existing intervention plans of promoting good dietary practices and overall longevity among Malaysian adults. Hence, we aimed to examine the association between daily adequate intake of FVs and all-cause mortality, CVD mortality, and cancer mortality among Malaysian adults.

## 2. Materials and Methods

### 2.1. Study Design and Data Collection

This study utilized the data from the National Health & Morbidity Survey (NHMS) 2011 and mortality data from the National Registration Department (NRD) Malaysia. Briefly, the NHMS 2011 was a population-based cross-sectional survey (i.e., nationally representative data of non-institutionalized Malaysian population) conducted to determine the morbidity and health status of Malaysian adults aged 18 years and above. The data from approximately 18,300 individuals who participated in the NHMS 2011 survey were retrieved. The NHMS used a two-stage stratified sampling design to select a nationally representative sample. The Malaysian population was first stratified by states, followed by further stratification by urban and rural residences.

The primary sampling units were enumeration blocks (EBs) provided by the Malaysian Department of Statistics (DOSM) from the 2010 census, where a total of 794 EBs (i.e., 484 urban and 310 rural) were systematically selected from all the Malaysian EBs (about 75,000) via a probability-proportional-to-size sampling technique. Then, the living quarters (LQs) represent the secondary sampling units, where a total of twelve LQs were randomly selected from the approximately 80 to 120 LQs in each EB. Finally, all households and eligible household members in the selected LQs were included in the sample and were invited to participate in the NHMS 2011 survey. The eligible respondents who agreed to participate provided their written consent prior to the interview for the survey data collection. The study protocol of the NHMS 2011 was approved by the Malaysian Medical and Research Ethics Committee (MREC), Ministry of Health Malaysia (NMRR-10-757-6837). The detailed survey methodology was described elsewhere [17].

The NHMS 2011 survey data collection was performed between April and July 2011. It was conducted by the trained interviewers via face-to-face interviews using standardized pre-validated structured questionnaires. Information on self-rated health (SRH), socio-demographic characteristics, lifestyle, and self-reported medically diagnosed chronic diseases was collected during the interview. A total of 18,231 eligible adults aged 18 years and older living in the sampled households were interviewed.

### 2.2. Assessment of Dietary Practice

To assess the dietary practices of the participants, data about the daily intake of fruit, vegetables, plain water, and sweetened sugary beverages and habits of breakfast and heavy meals after dinner were collected using the food frequency questionnaire (FFQ) [17]. In the present study, we focused on the collected data on the daily intake of FVs. Participants were requested to answer four questions to assess the daily intake of FVs by the participants within a seven-day period. The first question (Q1) was, “In a typical week, how many days do you eat fruit?” Following that, the respondents were asked to answer the second to the fourth questions: Q2: “On the day that you usually eat fruit, how much did you eat?”; Q3: “In a typical week, on how many days do you eat vegetables?”; Q4: “On the day that you usually eat vegetables, how much did you eat on that day?” During the interview, the food photo-elicitation technique was utilized as a tool to help the participants recall the serving size of FVs taken daily, where photographs were used to depict a single serving of commonly consumed FVs. For example, one cup of papaya or one medium-sized apple was considered a single serving of fruit, whilst one cup of chopped raw leafy green vegetables or a half cup of other vegetables (cooked or chopped raw) was considered a single serving of vegetables.

The responses to the first and second questions were multiplied, and the 7-day average was calculated to determine the total number of fruit servings per day, where ≥2 servings per day of fruit is defined as adequate; otherwise, it is inadequate. Correspondingly, we determined the total number of vegetable servings per day using Q3 and Q4, where the cut-off to define the adequate intake of vegetable servings per day was ≥3 servings per day. Referring to the Malaysian Dietitian Guidelines 2020, an individual is classified as having adequate daily FV intake when ≥5 servings of fruit and vegetables (i.e., ≥2 servings of fruit and ≥3 servings of vegetables) were consumed daily; otherwise, it is inadequate [3]. Adequate fruit intake was defined as consuming ≥2 servings of fruit per day, whilst adequate vegetable intake was defined as consuming ≥3 servings of vegetables per day [3].

### 2.3. Covariates

In this study, there were three categories of covariates, i.e., sociodemographic characteristics, lifestyle risk factors, and health conditions. The covariates that were categorized as sociodemographic characteristics were gender, age, ethnicity, residential area, marital status, educational level, and monthly household income, whilst obesity, alcohol consumption, smoking status, and physical activity were covariates for lifestyle risk factors. As for the health conditions, the covariates that were included were diabetes, hypertension, and hypercholesterolemia.

All participants aged 18 years old and above were categorized into three groups: 18–39 years, 40–59 years, and 60 years and above. The participants self-reported their ethnicity and were classified into four subcategories: (i) Malay, (ii) Chinese, (iii) Indian, and (v) others. The residential area was categorized into urban and rural areas. As for the educational level, the classification was based on the local educational system: (i) no formal education, (ii) primary education, (iii) secondary education, and (iv) tertiary education. The total monthly household income was categorized using the Malaysian household income classification published by the DOSM as follows: (i) bottom 40% (B40) with monthly income below RM 4850 (i.e., USD 1029), (ii) middle 40% (M40) with monthly income range between RM 4851 and RM 10,970 (i.e., between USD 1030 and USD 2328), (iii) and top 20% (T20) with monthly income above RM 10,971 (above USD 2328). The total monthly household incomes are henceforth reported in USD, and values in USD were based on the currency exchange rate on 24 June 2024.

The lifestyle factors that we have included in the present study were body mass index (BMI), alcohol intake, smoking status, and physical activity. Based on the World Health Organization (WHO) cut-off, the BMI was categorized as (i) underweight (BMI below 18.5), (ii) normal (BMI ranging between 18.5 and 24.9), (iii) overweight (BMI ranging between 25.0 and 29.9), and (iv) obese (BMI above 30). Alcohol intake status was categorized as (i) never and (ii) ever (current and former drinker). As for the smoking status, there were three categories: (i) never, (ii) former smoker, and (iii) current smoker. To assess the level of physical activity of the respondents, the short version of the International Physical Activity Questionnaire was used, where the physical activity was categorized as (i) inactive and (ii) active.

The health conditions of the participants, such as diabetes, hypertension, and hypercholesterolemia, were either self-reported based on medical history or defined based on the measurement of blood fasting sugar, cholesterol levels, and blood pressure during recruitment. Firstly, a respondent who was clinically diagnosed with diabetes and/or had a fasting capillary blood glucose of 6.1 mmol/L or more (or non-fasting blood glucose of more than 11.1 mmol/L) was classified as diabetes. Secondly, a respondent who was clinically diagnosed with hypertension and/or had a systolic blood pressure of 140 mmHg or more and/or diastolic blood pressure of 90 mmHg or more was classified as having hypertension. Lastly, a respondent who was clinically diagnosed with hypercholesterolemia and/or had a total blood cholesterol of 5.2 mmol/L or more was classified as having hypercholesterolemia.

### 2.4. Mortality Follow-Up

The participants of the NHMS 2011 were followed up for approximately 11 years, from April 2011 until December 2021. Mortality data during this period were obtained by matching the survey participants’ identification numbers with records in the death register administered by the NRD Malaysia. The cause of death was classified using the International Classification of Diseases, Tenth Revision (ICD-10). The three main outcomes of interest of the present study were all-cause mortality, CVD mortality, and cancer mortality.

### 2.5. Statistical Analysis

The statistical analyses were performed using the IBM SPSS statistical packages version 29.0 with a complex samples add-on module (IBM Corp., Armonk, NY, USA). All planned analyses were performed using the complex survey design and unequal selection probabilities. The prevalence of sociodemographic characteristics, lifestyle risk factors, and health conditions were estimated and presented as frequencies, percentages, and 95% confidence intervals (CIs) by the adequacy of FV daily intake status.

The multivariable analysis by Cox regression was carried out to determine the association between daily adequate FV intake status and all-cause, CVD, and cancer mortalities. The Cox proportional-hazards regression models were used to determine the associations between daily adequate FV intake status and all-cause, CVD, and cancer mortalities, i.e., Model 1 was adjusted for age and gender, and Model 2 was adjusted for sociodemographic and lifestyle factors and health conditions. For sensitivity analyses, Cox regression models were estimated for mortality by excluding mortalities in the first two years of follow-up. The overall proportional hazards test was performed for all the models to check assumption violations of the complex sample Cox regression model. A *p*-value less than 0.05 violated the assumption. Furthermore, the unparallel patterns on the log minus log plot of the survival function test indicated proportionate hazards assumption was violated. Two-way interactions between all the independent variables in each model were explored, and no significant two-way interactions were observed (*p* > 0.05). The figures were generated using GraphPad Prism version 5.01 for Windows (GraphPad Software, Boston, MA, USA).

## 3. Results

At the baseline, the mean age of the participants of this study was 39.00 (±0.22) years old. A total of 1809 deaths were recorded, i.e., 9.9% of the study population (n = 18,211), during the follow-up of 10.9 years. Of these, 374 and 216 were CVD and cancer mortalities, whilst the remaining 1219 deaths were other causes of death (Appendix A). Table 1 shows the baseline sociodemographic characteristics of the study participants. Among the study population, 51.1% were males, 49.8% were Malays, 56.8% were aged between 18 and 39 years old, 73.1% resided in urban areas, 65.4% were married, 47.2% attended secondary education, and 75.1% were in the B40 income group. The analysis of the lifestyle factors of the study population showed that 47.2% had normal BMI, 80.4% were non-drinkers, 68.5% never smoked, and 64.9% were physically active. Further analysis of the health conditions of the study population demonstrated that 15.2% were medically diagnosed with diabetes, 32.6% with hypertension, and 35.1% with hypercholesterolemia. The prevalence of the daily adequate FV intake, fruit intake, and vegetable intake are presented in Figure 1. At baseline, the prevalence of daily adequate intake of FVs was relatively low, with 8.8% of the study population having ≥5 servings of FVs. With further stratification by fruit and vegetable categories, we observed 14.6% of the study population had daily adequate fruit intake, whilst 13.7% had daily adequate vegetable intake.

A comparison of the characteristics of the study population via stratification of the adequacy status of daily FV intake, daily fruit intake, and daily vegetable intake is shown in Table 2. Those with daily adequate FV intake were generally Chinese, aged between 40 and 59 years old, married, receiving tertiary education, and physically active. Meanwhile, those with daily adequate fruit intake were generally female, Chinese, aged between 40 and 59 years old, urban dwellers, married, tertiary education recipients, in the T20 income group, overweight, former smokers, physically active, and being diagnosed with health conditions, i.e., diabetes, hypertension, and hypercholesterolemia. As for vegetable intake, those having daily adequate intake were Chinese, married, tertiary education recipients, obese, and physically active. The prevalence of all-cause mortality among those with daily adequate FV intake was 7.3%, whilst 1.5% and 1.3% for the respective CVD and cancer mortalities (Figure 2A). We further observed that the prevalence of all-cause mortality, CVD mortality, and cancer mortality among Malaysian adults with adequate fruit intake only were 7.3%, 1.8%, and 1.3%, respectively (Figure 2B). Among those with daily adequate vegetable intake only, the prevalences of all-cause mortality, CVD mortality, and cancer mortality were 7.7%, 1.5%, and 1.3%, respectively (Figure 2C).

In our fully adjusted model, daily adequate FV intake was not associated with all-cause mortality (aHR:1.01, 95% CI: 0.79–1.31), CVD mortality (aHR:0.91, 95% CI: 0.57–1.47), and cancer mortality (aHR:1.27, 95% CI: 0.74–2.17) (Figure 3a). We further performed the sensitivity analysis by excluding the death observed in the first two years of the baseline. Our results demonstrated that no association between daily adequate FV intake and all-cause mortality (aHR:1.03, 95% CI: 0.78–1.36), CVD mortality (aHR:1.00, 95% CI: 0.61–1.67), and cancer mortality (aHR:1.21, 95% CI: 0.65–2.23), after adjusting for sociodemographic, lifestyle risk factors, and health conditions (Figure 3b). Following subsequent analyses, we stratified fruit and vegetables as separate independent variables. Our results demonstrated no significant association between adequate vegetable intake and all-cause mortality (aHR:1.19, 95% CI: 0.88–1.61), CVD mortality (aHR:0.81, 95% CI: 0.40–1.63), and cancer mortality (aHR:0.98, 95% CI:0.41–2.34), even after excluding the deaths occurred in the first two years of observation (Appendix A). Likewise, no association was observed between adequate fruit intake and all-cause mortality (aHR: 0.84, 95% CI: 0.68–1.03), CVD mortality (aHR:0.93, 95% CI: 0.57–1.53), and cancer mortality (aHR:0.90, 95% CI: 0.59–1.37), even after excluding the deaths that occurred in the first two years of observation (Appendix A).

## 4. Discussion

In the present study, we investigated adequate FV intake and its association with all-cause, CVD, and cancer mortalities among Malaysian adults. Our findings showed a relatively low prevalence of daily adequate FV intake, fruit intake, and vegetable intake among Malaysian adults aged 18 years and above. Following further investigation, no significant associations between daily adequate intakes of FVs, fruit, and vegetables with lower risk of all-cause, CVD, and cancer mortalities among Malaysian adults were observed.

Several local studies showed a low prevalence of daily FV intake among Malaysians. For instance, the national survey conducted in 2015 reported an alarming low prevalence of adequate FV intake, i.e., 2.8% among Malaysian adults [6]. Another study within the Federal Territory of Kuala Lumpur among the B40 Malaysian adults (with monthly income below USD 1029) reported that 10.2% of men and 10.8% of women were found to consume adequate daily servings of FVs [18]. A later cross-sectional study conducted using the online platform demonstrated a sharp decrease, where only 0.2% of Malaysian adults achieved the recommended five servings per day during the COVID-19 pandemic, which could be due to the strict movement control order implemented by the federal government of Malaysia in response to the pandemic [19]. These reported local findings were much lower compared to our neighboring countries. For instance, the Thailand National Health Examination Survey III reported that 25% of Thais consumed adequate FVs [20]. In another nationally representative study involving 14,706 Vietnamese participants aged between 25 and 64 years old, it was reported that approximately 20% of the population have more than five servings of FVs daily in a typical week [21].

Findings from other regions outside Asia demonstrated the reported prevalence of FV intake in their populations was higher compared to our present findings. For example, 5.0% of the Austrian adults were reported to consume adequate FV intake [22]. Native Portuguese have an 18.5% prevalence of adequate FV intake [23]. In Australia, the AusDiab Study involving 8966 participants at baseline reported that 24.1% of Australians had adequate FV intake [24]. Quadir and Akhtar-Danesh reported that an estimated 33.5% of the overall Canadian population consumed five or more daily servings of FVs [25]. However, further stratification by racial groups showed a difference in the proportion of racial groups having an adequate daily intake of FVs, where the highest proportion of 39% was reported in Latin America, whilst Southeast Asia has the lowest proportion at 13%. Putting this all together, the different prevalences of adequate daily FV intake reported across different regions on the globe indicated that different cultural practices across the different ethnic groups influence their dietary patterns, including the frequency of FV intake. Another plausible explanation for this observed pattern is the affordability of FVs. The Prospective Urban Rural Epidemiology (PURE) study, which enrolled participants from communities in 18 countries, reported that individuals in countries with low gross national income consume fewer FVs and spend a greater proportion of their income purchasing food than those in high-income countries [26]. Furthermore, the regional difference in the accessibility and availability of FVs may also affect the dietary patterns that lead to a low prevalence of daily adequate FV intake, which is pending further investigation.

In the present study, we observed no significant association between daily adequate FV intake and lower risk of all-cause, CVD, and cancer mortalities among Malaysian adults. There are several plausible explanations for the observed non-significant findings. To begin with, the employed study design in this present study, a retrospective cross-sectional study, could be one of the possible explanations for the mentioned non-significant findings. At recruitment, the daily FV intake of the participants was measured based on a single measurement on a weekly average instead of collecting the prospective daily FV intake. Additionally, there could be a potential deviation from the self-report that arises from the recall period or recall bias at recruitment. Furthermore, dietary patterns tend to differ based on gender, culture, ethnic background, and socioeconomic status. Moreover, a dietary pattern is subject to significant alterations over time, resulting in changes in FV intake. Hence, the collected daily adequate FV intake data could not reflect the relevant lifelong intake of FVs among Malaysian adults. A well-designed national representative prospective study in the future is needed to examine the effects of FV intake on long-term health outcomes, including longevity, among Malaysian adults.

Next, no available data on the category or type of vegetables may be another plausible explanation for the observed non-significant findings in our study. Legume intake alone has been reported to be associated with a lower risk of mortality. For instance, a prospective cohort study involving 135,335 individuals aged between 35 and 70 years old in 18 countries across seven geographical regions reported legume intake was associated with 18% lower risk of non-cardiovascular death and 26% lower risk of all-cause death when taking more than one serving per day [27]. A study in a Chinese population reported a similar finding, where legume intake was associated with a 6% lower risk of all-cause mortality [28]. A recent large prospective cohort comprising more than 480,000 US men and women aged between 50 and 71 years old showed that specific botanical groups of FV intake (i.e., lettuce and the cruciferous family) were associated with liver cancer mortality [29], suggesting the need to investigate the different botanical groups of FVs and mortality risk. In the present study, the data on legume intake were not collected separately but instead grouped into vegetable intake as a whole. Moreover, we only assessed the number of daily servings of FVs and fruit and vegetables alone, and no further assessment of the group or type of vegetable intake, nutrient density, and variety of FVs consumed was performed in our study. Different FVs contain varying nutrient contents, which might influence the risk of CVD and cancer mortalities. Not all fruits and vegetables offer the same health benefits, and consuming a diverse range might be necessary to observe significant effects. Moreover, the overall dietary pattern might influence mortality outcomes more than individual fruit and vegetable intake alone [30,31]. Those who consume high amounts of fruit and vegetables might also consume other unhealthy foods, such as ultra-processed foods high in saturated fat, refined sugar and salt [32], sugar-sweetened beverages [33], red meat, and processed meat, which could confound the results. Therefore, a future prospective study using 3-day dietary records or food frequency intake alongside assessment of food quality scores to evaluate the micronutrient profiles of various types of FVs consumed [34,35]. Data from such a study could provide more insights into the causal relationship between FV intake and mortality.

In addition, the mode of preparation of the FVs that were not examined in the present study could also be one of the explanations for the observed non-significant findings. Evidence from the European Prospective Investigation into Cancer and Nutrition (EPIC) Study comprising more than 450,000 participants of European descendants from 10 countries reported the inverse association with risk of mortality was more pronounced for high raw vegetable intake (i.e., 16% lower risk of mortality) when compared to the cooked vegetable intake (i.e., 6% lower risk of mortality) [13]. Later, the findings from the same EPIC study reported that high raw vegetable intake was associated with a lower risk of death from respiratory, neoplasm, and mental diseases and behavioral disorders [12]. Evidence from the study by Miller and colleagues was concorded with the EPIC study, where raw vegetable intake was inversely associated with mortality, in particular, cardiovascular mortality [27]. Cooked vegetables alter the nutrients’ bioavailability (i.e., phytochemicals, vitamins, minerals, and fiber), structure, digestibility, and destruction of the digestive enzymes, which may affect the health outcomes of the consumers [36,37]. Nevertheless, although the mode of preparation of the FVs was not examined in the present study, cooked vegetables are commonly presented on dining tables, albeit Malaysia is known for cultural diversity. Future studies on the nutritional values of cooked vegetables and their impact on health outcomes in the future will not only aid consumers with healthy dietary practices but also the policymakers in improvising the existing efforts to raise awareness of the importance of healthy eating practices.

Following that, the insufficient observation/follow-up period may also contribute to the observed non-significant findings for cancer mortality. Findings from different studies showed that the association between adequate or high intake of FVs and cancer mortality was inconsistent [13,14,28,29,38,39]. In the present study, we did not observe any significant association between daily adequate intake and risk of cancer mortality among Malaysian adults. This could be due to the longer induction periods that exist for cancer mortality [13]. A prospective longitudinal cohort study with a longer period of observation to investigate the influences of specific types of FVs on cause-specific mortality is needed. Findings from this kind of study will benefit consumers in terms of the dimensions of food choice and further influence the dietary patterns of the consumers.

Lastly, the competing risks, such as death without CVD or cancer prior to these outcomes, were not addressed in this present study. Addressing the competing risks is important to avoid violating key assumptions in survival analysis. Hence, future studies in this context should employ alternative statistical methods that properly account for competing risks, such as the sub-distribution hazard model (i.e., the Fine and Gray model) [40]. Employing the Fine and Gray model to estimate survival probabilities for outcomes, such as all-cause mortality, CVD mortality, and cancer mortality, by considering the influence of competing risks will provide a more accurate hazard ratio. Furthermore, the event–time assumptions and censoring processes will be correctly handled when this model is employed. Such adjustment will enhance the accuracy and interpretability of results in future related studies.

## 5. Conclusions

In conclusion, no significant associations were observed between daily adequate FV intake with risk of all-cause, CVD, and cancer mortalities among Malaysian adults. There is a need for future studies to investigate the effects of the types of FV intake and their preparation methods on the risk of mortality. These findings will provide a holistic insight into the causal relationship between FV intake and mortality.

## Figures and Tables

**Figure 1 nutrients-16-03200-f001:**
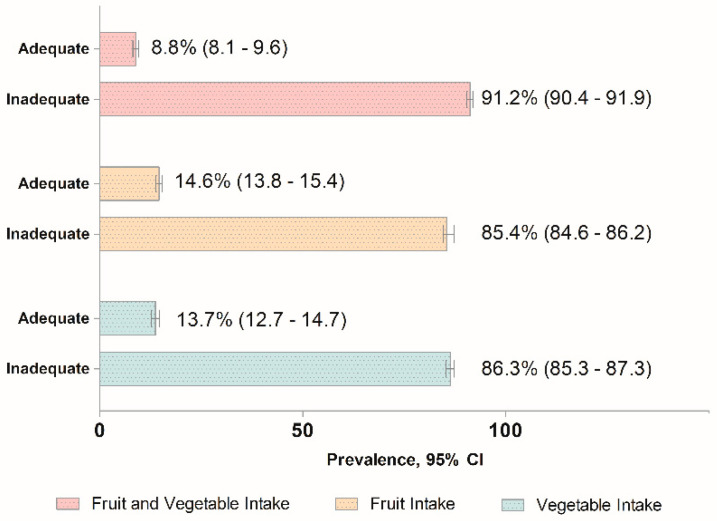
The prevalence of adequate daily intake of FVs and fruit and vegetable intakes among the Malaysian adults aged 18 years and above.

**Figure 2 nutrients-16-03200-f002:**
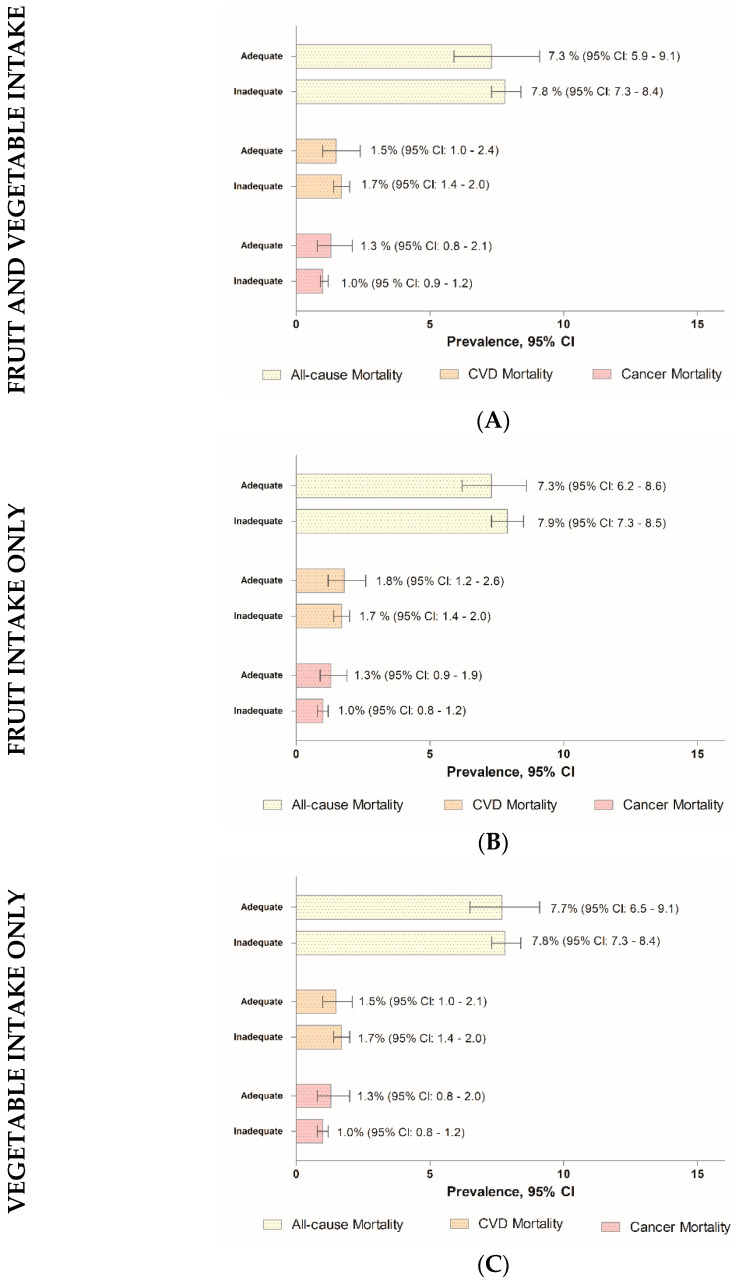
The prevalence of all-cause, CVD, and cancer mortalities among the Malaysian adults aged 18 years and above. (**A**) Fruit and vegetable intake, (**B**) fruit intake only, and (**C**) vegetable intake only.

**Figure 3 nutrients-16-03200-f003:**
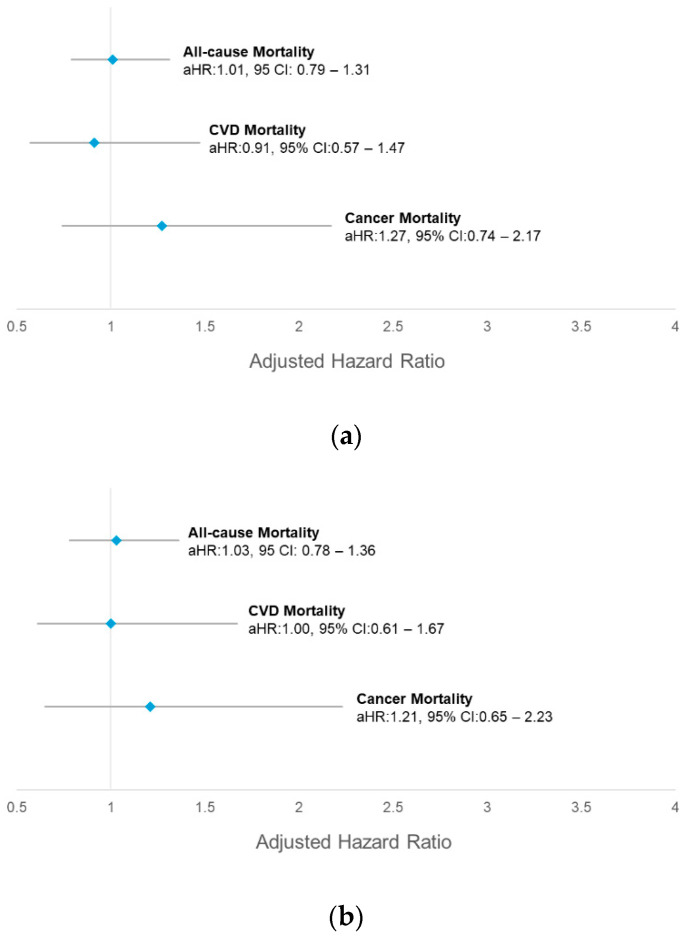
Associations between daily adequate FV intake with all-cause mortality, CVD mortality, and cancer mortality among the NHMS 2011 respondents. (**a**) Multiple Cox regression adjusted for sex, ethnicity, age, residential area, marital status, education level, monthly household income, general obesity, alcohol intake, smoking, physical activity, diabetes, hypertension, and hypercholesterolemia. (**b**) Multiple Cox regression adjusted for sex, ethnicity, age, residential area, marital status, education level, monthly household income, general obesity, alcohol intake, smoking, physical activity, diabetes, hypertension, and hypercholesterolemia after excluding mortality in the first 2 years. CVD = cardiovascular disease; aHR = adjusted hazard ratio; 95% CI = 95% confidence interval.

**Table 1 nutrients-16-03200-t001:** The characteristics of the NHMS 2011 respondents aged 18 years and above (n = 18,211).

Characteristic	Estimated Population	Participant	
		Count (n)	% (95% CI)
*Sociodemographic*			
**Gender**			
Male	9,092,748	8527	51.1 (50.2–52.0)
Female	8,713,236	9684	48.9 (48.0–49.8)
**Ethnicity**			
Malay	8,859,013	10,377	49.8 (46.9–52.6)
Chinese	4,528,821	3519	25.4 (23.0–28.0)
Indian	1,252,004	1458	7.0 (5.9–8.3)
Others	3,166,146	2857	17.8 (16.0–19.7)
**Age in the group (years old)**			
18–39	10,115,010	8675	56.8 (55.5–58.1)
40–59	5,630,963	6782	31.6 (30.6–32.7)
≥60	2,060,011	2754	11.6 (10.8–12.4)
**Residential area**			
Urban	13,021,370	10,595	73.1 (71.9–74.3)
Rural	4,784,614	7616	26.9 (25.7–28.1)
**Marital status**			
Single	5,083,177	4230	28.6 (27.3–29.9)
Married	11,633,350	12,492	65.4 (64.1–66.7)
Widow(er)/divorcee	1,069,296	1467	6.0 (5.6–6.5)
**Education level**			
No formal education	1,080,059	1445	6.1 (5.5–6.7)
Primary education	3,812,473	4360	21.4 (20.3–22.6)
Secondary education	8,395,931	8278	47.2 (45.8–48.5)
Tertiary education	4,244,311	3863	23.8 (22.4–25.3)
Unknown status	273,209	265	1.5 (1.3–1.9)
**Monthly household income**			
B40	13,369,330	14,127	75.1 (73.1–77.0)
M40	3,704,696	3468	20.8 (19.3–22.4)
T20	731,959	616	4.1 (3.3–5.2)
** *Lifestyle factor* **			
**Obesity**			
Underweight	1,360,608	1277	8.3 (7.7–9.0)
Normal	7,705,381	7622	47.2 (46.0–48.4)
Overweight	4,795,532	5148	29.4 (28.4–30.4)
Obese	2,462,152	2750	15.1 (14.3–15.9)
**Alcohol consumption**			
Never	14,213,412	15,417	80.4 (78.8–81.8)
Ever	3,475,720	2675	19.6 (18.2–21.2)
**Smoking**			
Never	12,058,065	12,686	68.5 (67.4–69.5)
Former	1,157,210	1237	6.6 (6.1–7.1)
Current	4,396,934	4131	25.0 (24.0–26.0)
**Physical Activity**			
Inactive	6,199,601	6464	35.1 (33.9–36.4)
Active	11,446,310	11,586	64.9 (63.6–66.1)
** *Health conditions* **			
Diabetes	2,618,580	3196	15.2 (14.3–16.1)
Hypertension	5,771,332	6690	32.6 (31.6–33.7)
Hypercholesterolemia	6,162,913	7002	35.1 (33.9–36.2)

NHMS = National Health and Morbidity Survey; B40 = bottom 40% of income earners; M40 = middle 40% of income earners; T20 = top 20% of income earners.

**Table 2 nutrients-16-03200-t002:** Adequate daily FV, fruit, and vegetable intakes by sociodemographic characteristics, lifestyle factors, and health status among the NHMS 2011 respondents.

Characteristic	Fruit and Vegetable Intake (n = 17,971)					Fruit Intake (n = 18,029)					Vegetable Intake (n = 18,112)				
	Adequate		Inadequate		*p*-Value	Adequate		Inadequate		*p*-Value	Adequate		Inadequate		*p*-Value
	n	%, 95% CI	n	%, 95% CI		n	%, 95% CI	n	%, 95% CI		n	%, 95% CI	n	%, 95% CI	
** *Sociodemographic* **															
**Gender**					0.747					<0.001					0.775
Male	764	8.7 (7.9–9.6)	7645	91.3 (90.4–92.1)		1171	13.3 (12.3–14.4)	7276	86.7 (85.6–87.7)		1134	13.8 (12.6–15.0)	7359	86.2 (85.0–87.4)	
Female	785	8.9 (8.0–9.9)	8777	91.1 (90.1–92.0)		1576	15.9 (14.9–17.0)	7988	84.1 (83.0–85.1)		1207	13.6 (12.4–14.8)	8447	86.4 (85.2–87.6)	
**Ethnicity**					<0.001					<0.001					0.007
Malay	894	9.3 (8.4–10.3)	9348	90.7 (89.7–91.6)		1578	14.7 (13.8–15.8)	8668	85.3 (84.2–86.2)		1223	12.9 (11.9–14.0)	9115	87.1 (86.0–88.1)	
Chinese	331	9.8 (8.4–11.5)	3149	90.2 (88.5–91.6)		660	18.1 (16.2–20.3)	2824	81.9 (79.7–83.8)		556	16.0 (14.1–18.1)	2950	84.0 (81.9–85.9)	
Indian	81	4.6 (3.5–6.0)	1364	95.4 (94.0–96.5)		17	11.2 (9.1–13.7)	1271	88.8 (86.3–90.9)		163	10.9 (8.8–13.5)	1292	89.1 (86.5–91.2)	
Others	243	7.6 (6.3–9.1)	2561	92.4 (90.9–93.7)		334	10.5 (9.0–12.1)	2501	89.5 (87.9–91.0)		399	13.6 (11.5–16.1)	2449	86.4 (83.9–88.5)	
**Age groups (years old)**					0.042					<0.001					0.158
18–39	691	8.5 (7.6–9.4)	7856	91.5 (90.6–92.4)		1046	11.5 (10.6–12.4)	7529	88.5 (87.6–89.4)		1103	13.5 (12.4–14.7)	7544	86.5 (85.3–87.6)	
40–59	641	9.7 (8.7–10.9)	6061	90.3 (89.1–91.3)		1251	19.2 (17.8–20.6)	5457	80.8 (79.4–82.2)		902	14.4–13.1–15.8)	5852	85.6 (84.2–86.9)	
≥60	217	7.9 (6.6–9.4)	2505	92.1 (90.6–93.4)		450	17.4 (15.4–19.6)	2278	82.6 (80.4–84.6)		336	12.4 (10.8–14.2)	2410	87.6 (85.8–89.2)	
**Residential area**					0.452					<0.001					0.471
Urban	926	8.9 (8.0–9.9)	9538	91.1 (90.1–92.0)		1719	15.5 (14.5–16.6)	8759	84.5 (93.4–85.5)		1423	13.9 (12.7–15.1)	9140	86.1 (84.9–87.3)	
Rural	623	8.4 (7.5–9.5)	6884	91.6 (90.5–92.5)		1028	12.1 (11.0–13.2)	6505	87.9 (86.8–89.0)		918	13.1 (11.7–14.8)	6666	86.9 (85.2–88.3)	
**Marital status**					0.002					<0.001					0.001
Single	298	7.8 (6.7–9.0)	3860	92.2 (91.0–93.3)		450	10.2 (9.1–11.4)	3728	89.8 (88.6–90.9)		481	12.4 (10.9–14.0)	3740	87.6 (86.0–89.1)	
Married	1157	9.5 (8.6–10.4)	11,202	90.5 (89.6–91.4)		2070	16.3 (15.3–17.3)	10,310	83.7 (82.7–84.7)		1707	14.6 (13.5–15.7)	10,746	85.4 (84.3–86.5)	
Widow(er)/divorcee	94	6.5 (5.0–8.4)	1349	93.5 (91.6–95.0)		225	16.8 (14.2–19.9)	1224	83.2 (80.1–85.8)		153	10.4 (8.5–12.7)	1309	89.6 (87.3–91.5)	
**Educational level**					0.005					<0.001					0.017
No formal	89	5.3 (4.1–6.9)	1331	94.7 (93.1–95.9)		146	9.0 (7.3–11.1)	1284	91.0 (88.9–92.7)		159	9.8 (7.9–12.2)	1278	90.2 (87.8–92.1)	
Primary	363	8.1 (7.0–9.4)	3939	91.9 (90.6–93.0)		602	12.7 (11.4–14.2)	3713	87.3 (85.8–88.6)		571	14.2 (12.5–16.0)	3780	85.8 (84.0–87.5)	
Secondary	715	9.0 (8.1–9.9)	7472	91.0 (90.1–91.9)		1289	14.8 (13.7–15.9)	6909	85.2 (84.1–86.3)		1035	13.3 (12.2–14.5)	7218	86.7 (85.5–87.8)	
Tertiary	368	10.0 (8.7–11.6)	3452	90.0 (88.4–91.3)		683	17.4 (15.7–19.3)	3150	82.6 (80.7–84.3)		547	15.0 (13.4–16.7)	3308	85.0 (83.3–86.6)	
Unclassified	14	6.7 (3.1–13.7)	228	93.3 (86.3–96.9)		27	11.5 (6.9–11.5)	208	88.5 (81.5–93.1)		29	12.7 (8.0–19.6)	222	87.3 (80.4–92.0)	
**Monthly household income**					0.094					<0.001					0.085
B40	1159	8.5 (7.7–9.2)	12,776	91.5 (90.8–92.3)		1941	13.2 (12.4–14.1)	12,013	86.8 (85.9–87.6)		1754	13.3 (12.2–14.4)	12,318	86.7 (85.6–87.8)	
M40	322	9.5 (8.2–11.1)	3106	90.5 (88.9–91.8)		670	18.3 (16.5–20.2)	2780	81.7 (79.8–83.5)		484	14.5 (12.9–16.3)	2978	85.5 (83.7–87.1)	
T20	68	11.4 (8.1–15.8)	540	88.6 (84.2–91.9)		136	21.1 (15.9–27.5)	471	78.9 (72.5–84.1)		103	16.8 (13.3–21.1)	510	83.2 (78.9–86.7)	
** *Lifestyle factor* **															
**Obesity**					0.681					<0.001					0.015
Underweight	103	8.3 (6.5–10.5)	1151	91.7 (89.5–93.5)		146	9.8 (8.0–11.9)	1116	90.2 (88.1–92.0)		149	11.3 (9.3–13.5)	1126	88.7 (86.5–90.7)	
Normal	686	9.1 (8.1–10.1)	6825	90.9 (89.9–91.9)		1087	14.3 (13.1–15.5)	6481	85.7 (84.5–86.9)		1057	14.8 (13.5–16.1)	6547	85.2 (83.9–86.5)	
Overweight	427	8.6 (7.6–9.8)	4679	91.4 (90.2–92.4)		873	16.0 (14.7–17.4)	4226	84.0 (82.6–85.3)		645	13.1 (11.8–14.4)	4495	86.9 (85.6–88.2)	
Obese	244	9.5 (8.1–11.0)	2476	90.5 (89.0–91.9)		438	14.6 (13.0–16.3)	2295	85.4 (83.7–87.0)		349	14.0 (12.3–15.9)	2392	86.0 (84.1–87.7)	
**Alcohol intake**					0.976					0.081					0.081
Never	1320	8.8 (8.1–9.6)	13,923	91.2 (90.4–91.9)		2368	14.9 (14.1–15.8)	12,924	85.1 (84.2–85.9)		1943	13.3 (12.4–14.4)	13,437	86.7 (85.6–87.6)	
Ever	221	8.8 (7.4–10.5)	2410	91.2 (89.5–92.6)		371	13.3 (11.6–15.1)	2280	86.7 (84.9–88.4)		18	15.2 (13.2–17.5)	2279	84.8 (82.5–86.8)	
**Smoking**					0.489					<0.001					0.743
Never	1061	8.7 (7.9–9.5)	11,477	91.3 (90.5–92.1)		2042	15.7 (14.7–16.7)	10,512	84.3 (83.3–85.3)		1614	13.5 (12.5–14.7)	11,036	86.5 (85.3–87.5)	
Former	99	8.2 (6.4–10.5)	1119	91.8 (89.5–93.6)		20	17.3 (14.8–20.1)	1027	82.7 (79.9–85.2)		155	13.9 (11.4–16.8)	1076	86.1 (83.2–88.6)	
Current	378	9.4 (8.2–10.7)	3694	90.6 (89.3–91.8)		476	10.6 (9.4–12.0)	3621	89.4 (88.0–90.6)		557	14.1 (12.7–15.7)	3566	85.9 (84.3–87.3)	
**Physical activity**					<0.001					<0.001					<0.001
Inactive	463	7.3 (6.4–8.3)	5899	92.7 (91.7–93.6)		83	12.5 (11.4–13.6)	5546	87.5 (86.4–88.6)		728	11.5 (10.4–12.7)	5716	88.5 (87.3–89.6)	
Active	1083	9.7 (8.8–10.6)	10,386	90.3 (89.4–91.2)		1901	15.8 (14.7–16.8)	9626	84.2 (83.2–85.3)		1608	14.9 (13.8–16.1)	9954	85.1 (83.9–86.2)	
** *Health conditions* **															
Diabetes	273	8.4 (7.2–9.8)	2894	91.6 (90.–92.8)	0.627	524	16.5 (14.7–18.5)	2637	83.5 (81.5–85.3)	0.013	391	12.5 (11.0–14.1)	2795	87.5 (85.9–89.0)	0.12
Hypertension	578	8.8 (7.9–9.9)	6033	91.2 (90.1–92.1)	0.957	109	16.5 (15.2–17.8)	5546	83.5 (82.2–84.8)	<0.001	848	13.4 (12.1–14.7)	5821	86.6 (85.3–87.9)	0.512
Hypercholesterolemia	592	8.8 (7.8–9.8)	6333	91.2 (90.2–92.2)	0.882	1163	16.1 (14.9–17.3)	5786	83.9 (82.7–85.1)	0.002	865	12.8 (11.7–14.1)	6116	87.2 (85.9–88.3)	0.066

n = unweighted count, % = prevalence, 95% CI = 95% confidence interval.

## Data Availability

All the generated data from the present study is presented in this published article and its Appendix A. The data used for this study are not available publicly due to data protection policy. The data are, however, available upon reasonable request from the Sector for Biostatistics and Data Repository, Office of NIH Manager, National Institutes of Health Malaysia, with permission from the Director General of Ministry of Health Malaysia.

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
