# Peer review of "Daily Adequate Intake of Fruit and Vegetables and All-Cause, Cardiovascular Disease, and Cancer Mortalities in Malaysian Population: A Retrospective Cohort Study"

_nutrients, 2024, doi:10.3390/nu16183200_

Round 1

Reviewer 1 Report

Comments and Suggestions for Authors

The manuscript deals with a very important epidemiological and public health question. In general the manuscript is sound and methods appropriate. However, I still have some methodological remarks tthat needs to be addressed:

(1) The biggest issue is that the authors did not consider competing risks. Death w/o CVD or cancer prior to CVD or cancer death is a competing risk preventing the event of interest. Furthermore, if competing risk occur the event-time for the event of interest becomes infinite. That violates the assumption of non-informative censoring and that censored patients have the same progression as those where the event of interest occured. That needs to be appropriately addressed. Presented HR are just cause-specific HRs which only translate 1:1 to changes in survival probabilities and are thus vaild only in a world where the competing risk cannot happen.

(2) Conceptually it makes not sense to run a age/sex adjusted Cox and a second one which is fully adjusted. Due to the non-collapsability property of the hazard ratio this approach is flawed and does not tell anything even if the HR would change. So please present only the fully adjusted model.

(3) Furthermore, please include a DAG that give the rationale for the causal assumptions done that justify the adjustments for confounding.

(4) Since you only consider >=2 fruits + >= 3 vegetable as adequate the inadequate group is very heterogenuous and you throw away a lot of information.

Furthermore, you are dealing with a 2x2 exposure (>=2 fruits vs <2) and (>=3 vegetable vs <3) and the combination of that, this 2x2 exposure should be addressed by including both main effects and an interaction term in the Cox model. Furthermore, the authors should think about running the same approach but including the continuous values of both exposures into the models instead of the pre-categorized ones to gain more inside and use the full information.

(5) Methods need statements how models assumptions of the Cox model were checked.

Reviewer 2 Report

Comments and Suggestions for Authors

Lay Kim TAN and colleagues did a retrospective cohort study to investigate the effect of fruit and or vegetable intake on mortality in the Malaysian population.

The study is very well-designed and clearly presented. Especially the covariates are clearly defined. The manuscript is nicely written with clearly defined gaps and aim of the study. The methodology, sample collection, and participant characteristics are clearly defined.

The results section is fine and clear. The authors have analyzed the data rigorously to bring out the results.

Although the mortality was divided into 3 categories, but how was the mortality defined for all causes? Did you consider excluding the deaths due to accident or reasons other than sickness or old age?

The discussion is well-written and clearly compare the results with the previous studies. Authors have extensively discussed the caveats/ limitations of the study.

Why do authors consider results as negative rather than nonsignificant?

The conclusion reflects the aim.

Specific mistakes-

Line 126- Correct “toto” to “to”

Line 492- CHAPTER 1??? 

Comments on the Quality of English Language

English language is fine.
